# Monitoring plasma nucleosome concentrations to measure disease response and progression in dogs with hematopoietic malignancies

Heather Wilson-Robles[1,2]*, Emma Warry[1], Tasha Miller[1], Jill Jarvis[1], Matthew Matsushita[1], Pamela Miller[1], Marielle Herzog[3], Jean-Valery Turatsinze[3], Theresa K. Kelly[4], S. Thomas Butera[2], Gaetan Michel[3,4]

1 Small Animal Clinical Sciences Department, Texas A&M University, College Station, Texas, United States of America, 2 Volition Veterinary Diagnostics Development, Henderson, Nevada, United States of America, 3 Belgian Volition SRL, Parc Scientifique Crealys, Isnes, Belgium, 4 Volition America LLC, Henderson, Nevada, United States of America

* hwilson@cvm.tamu.edu

**Data Availability Statement:** All relevant data are within the manuscript and its Supporting information files.

## Abstract

### Background

Hematopoietic malignancies are extremely common in pet dogs and represent nearly 30% of the malignancies diagnosed in this population each year. Clinicians commonly use existing tools such as physical exam findings, radiographs, ultrasound and baseline blood work to monitor these patients for treatment response and remission. Circulating biomarkers, such as prostate specific antigen or carcinoembryonic antigen, can be useful tools for monitoring treatment response and remission status in human cancer patients. To date, there has a been a lack of useful circulating biomarkers available to veterinary oncology patients.

### Methods

Circulating plasma nucleosome concentrations were evaluated at diagnosis, throughout treatment and during remission monitoring for 40 dogs with lymphoma, acute myelogenous leukemia and multiple myeloma. Additionally, C-reactive protein and thymidine kinase-1 levels were recorded.

### Results

Plasma nucleosome concentrations were significantly higher at diagnosis and progressive disease than they were when dogs were in remission. All but two dogs had plasma nucleosome concentrations that returned to the low range during treatment. These two dogs had the shortest progression free and overall survival times. Dogs with the highest plasma nucleosome concentrations had a significantly shorter first progression free survival than dogs with lower plasma nucleosome concentrations at diagnosis. Plasma nucleosome concentrations correlated better with disease response and progression than either thymidine kinase or C reactive protein.

**Funding:** Funding for materials and author salaries was provided by Volition Veterinary Diagnostic Development. HWR and TM received salary from these funds. JVT, TK, MH, TB and GM are employees of Belgian. Volition & Volition America. The URL to the Belgian Volition website is: https://volition.com/. Some additional funding for HWR and TM salaries was provided by the Fred and Vola Palmer Chair of Comparative Oncology held by HWR.

**Competing interests:** JVT, TK, MH, TB and GM are employees of Belgian Volition, Volition Diagnostics UK ltd & Volition America, which have patents covering Nu. Q technology and are developers of Nu.QTM assays. Volition Veterinary is a joint venture between Belgian Volition and Texas A&M University. HWR is a paid consultant of Volition Veterinary. EW, TM, PM, MM and JJ have no conflicts of interest to declare. Additional salary support for TM was provided by the Fred and Vola Palmer Chair in Comparative Oncology held by HWR. The Palmers did not play a role in the study design, data collection and analysis, decision to publish, or preparation of the manuscript and only provided financial support for the authors' salaries (T.M.). This does not alter our adherence to PLoS One policies on sharing data and materials.

## Conclusions

Plasma nucleosome concentrations can be a useful tool for treatment monitoring and disease progression in dogs with hematopoietic malignancies.

## Introduction

Hematopoietic malignancies are common in pet dogs and represent a complex category of diseases arising from various types of blood cells, with lymphoma (LSA) being the most common of these malignancies in companion canines. According to Withrow and MacEwen's Small Animal Clinical Oncology, "the annual incidence of lymphoma has been estimated to be between 13–114 per 100,000 dogs at risk" [1].

Given the systemic nature of these tumors, chemotherapy is recommended as the first line of treatment in most cases. For most lymphomas and leukemias, CHOP chemotherapy is considered the standard of care in veterinary oncology. Traditional CHOP based protocols induce remission in 80–95% of canine LSA patients with an approximate 20–25% two year survival rate [1]. However, there is a subset of dogs that do not respond well to conventional protocols.

Remission monitoring and response to treatment in canine LSA has traditionally been determined using physical exam parameters and cytology. In some cases, serum calcium or globulin levels may also be used to monitor for progression and response, however these patients represent the minority of the cases managed. Liquid biopsy or circulating biomarker assays for the diagnosis and treatment monitoring of LSA in canines is an area of active investigation for several groups [2–6], though currently, there is no commercially available biomarker for treatment or remission monitoring in veterinary oncology.

Several blood-based biomarkers with utility in remission and treatment monitoring exist for human oncology patients including, PSA, CEA and CA-125 [7–11]. These markers not only have utility in screening for cancer in an 'at risk' population but are also used to monitor for treatment response and disease progression [7, 8, 10].

In humans, interim reassessment for LSA treatment monitoring frequently includes advanced imaging such as PET-CT scans with Deauville Scoring to assist with treatment decisions [12]. Surveillance imaging for asymptomatic patients in complete remission includes physical exams and routine blood work. Thoracic CT or chest radiographs may be done every 6–12 months for the first few years after therapy has been completed. Regular PET-CT scans are typically not recommended [12]. Biomarkers of treatment and remission monitoring in humans using circulating tumor DNA (ctDNA) is not yet common, but research is ongoing and has shown promise in a variety of studies [13–17].

In eukaryotic cells, eight histone proteins make up a nucleosome core providing the scaffolding upon which the DNA is wrapped and condensed into chromatin [18]. When cells die chromatin is decondensed and nucleosomes are released into circulation, typically as mononucleosomes. Longer polynucleosome fragments are released into circulation by specialized immune cells, called neutrophils, in response to infection in a process call NETosis, and thus circulating nucleosomes represent a dynamic portion of the cfDNA compartment [18]. Nucleosomes play important roles in nearly all major cellular functions including cellular proliferation, DNA repair, transcription, and phenotypical cellular identity. Circulating plasma nucleosomes maintain at least some epigenetic modifications once released into the blood stream [19–21]. Mononucleosomes can be cleared from the blood quickly, from minutes to hours depending on the quantity of nucleosomes in circulation, providing a real time glimpse into cellular turn over and death rates at any given time [21–23]. Plasma nucleosome levels

increase when the concentration within the blood exceeds the body's ability to clear them or when they bind to acute phase proteins [24].

Plasma nucleosome concentrations have been used as a tool to monitor the effectiveness of cytotoxic therapy for patients with lung, pancreatic and colon cancer as well as hematopoietic malignancies [25]. Significant decreases in these levels have been associated with remission status while increasing or persistently high levels have been associated with disease progression in patients receiving both chemotherapy and radiation therapy [24]. Though plasma nucleosomes are not tumor specific, the kinetics of plasma nucleosome concentrations can be valuable markers for therapeutic efficiency in a variety of human cancers and may allow for earlier adaption within treatment protocols [25].

Circulating plasma nucleosomes have been shown to be elevated in >75% of naïve canine LSA cases [2] and the utility of measuring plasma nucleosomes as a monitoring tool during treatment for a dog with hemangiosarcoma has also been published as a single case report [3]. Here, the authors describe the utility of measuring plasma nucleosome concentrations for monitoring treatment response and disease progression in a cohort of dogs with hematopoietic malignancies.

## Methods

Dogs presenting to the Texas A&M University (TAMU) Small Animal Oncology Service with newly diagnosed or newly relapsed hematopoietic malignancies were prospectively recruited to participate in this study. Eligibility criteria included a confirmed cytologic or histologic diagnosis of a hematopoietic malignancy, willingness to travel to TAMU for most chemotherapy and follow up visits and election of a definitive therapy protocol (chemotherapy for most patients). Dogs with elevations in kidney values were not eligible for this study. Dogs were allowed to have increases in liver values up to 1.5 times the upper limit of normal. Plasma was collected at diagnosis and at each subsequent chemotherapy or follow up visit until disease progression was noted. Disease progression was determined by physical examination using RECIST criteria and/or return of clinical signs and confirmed with cytology, imaging or both depending on the case [26]. If reinduction with another chemotherapy protocol was elected at progression, then patients were allowed to continue in the monitoring study. Plasma nucleosome concentrations as well as pertinent medical record data were recorded. Captured medical record data included visit dates, clinical diagnosis, phenotype, stage, age at diagnosis, breed, sex, body weight, date of diagnosis, date of first treatment, clinical response and date of disease progression. Hematology results, serum c-reactive protein (CRP) and thymidine kinase-1 (TK-1) levels were recorded when available.

The work presented in this study was approved by the TAMU Animal Care and Use Committee (AUP 2019–0211 and AUP 2017–0350) and all blood was collected with signed informed owner consent. Only client owned animals were recruited for this study, therefore, no animals were sacrificed as a result of this research. There were no painful procedures included as part of the study protocol, therefore, there were no specific anesthetic or analgesic protocols. There were no limitations on clinicians regarding the management of any painful conditions that evolved during the time the animals participated in the study.

All blood samples were collected from a jugular or peripheral vein using standard blood collection techniques. Samples were collected at diagnosis and at each chemotherapy and recheck visit thereafter until progressive disease was determined for a total of 803 samples collected. All dogs were fasted for a minimum of four hours before samples were collected. A total of 3–5 mL (depending on the size of the dog) was placed in K2-EDTA primed (lavender top) tubes (Becton, Dickinson and Company, Franklin Lakes, NJ). Samples were centrifuged

within one hour of collection at room temperature at 3000xg for 10 min [27]. Plasma was then immediately removed without disrupting the buffy coat layer, placed in pre-labeled cryovials and frozen at -80˚C to be run in batches.

## Nucleosome assay

All samples were tested using the Nu.Q® H3.1 assay (Belgian Volition, SRL, Isnes, Belgium). This enzyme-linked immunosorbent assay (ELISA) contains a capture antibody directed at histone 3.1 and a structurally specific nucleosome detection antibody [28]. Frozen samples were thawed and allowed to come to room temperature for 30 minutes prior to analysis. Assays were performed according to the manufacturer's protocol. Briefly, a standard curve was generated using the positive control stock provided. Samples were vortexed and centrifuged at 10,000 x g for two minutes at 4˚C. Samples were removed from the original tube, avoiding any pellet, and placed in a microcentrifuge tube. Twenty microliters of each sample were pipetted in triplicate into wells on 96 well plates. Next, 80μL of the assay buffer was added to each well. The plate was covered with sealing film and incubated on an orbital shaker at room temperature for 2.5 hours at 700 rpm. Plates were then decanted and washed three times using 200 μL washing buffer. Next, 100 μL of the detection antibody was added to each well, the plate was resealed and incubated at room temperature for 1.5 hours on the orbital shaker. The plates were then washed as described above. Streptavidin HRP conjugate was added and incubated for 30 min at room temperature in each well and washed before applying the colorimetric substrate solution and incubating the plates in the dark for 20 min. A stop solution was added to the wells and the plates were analyzed on a plate reader at 450 nm (BioTek Synergy H1 plate reader, BioTek Instruments, Winooski, VT). Results were expressed in optical density (OD), and the concentration were evaluated using a four-parameter logistic regression of a reference standard curve of known nucleosome concentration. If the %CV between the OD of the duplicate measurements was above 20%, the sample was repeated (Graphpad Software, version 9.0.0, Macintosh, GraphPad Software, San Diego, California USA, www.graphpad.com). Results are expressed in ng/ml. Categories of normal, grey zone and elevated nucleosomes are based on the commercial reference range for the H3.1 plasma nucleosome ELISA assay determined for frozen plasma samples from more than 250 healthy dogs and dogs with cancer (data not shown).

## CRP and thymidine kinase 1 assays

CRP and TK1 assays were performed on plasma at each of the data collection points. Specific time points, namely at enrollment, at best response and at the first documented progression event, were evaluated for all dogs in total. Samples were submitted to the TAMU Gastrointestinal Laboratory for plasma C-reactive protein (CRP) assays if sufficient sample quantity was present. If sample quantity was not sufficient for both nucleosome and CRP analysis, the nucleosome assay was given priority.

The Dog Thymidine Kinase ELISA Kit (TK1) (Abcam, Waltham, MA) was used to evaluate plasma TK1 levels. The assay was performed according to the manufacturer's protocol. Briefly, 100μl of sample was added to wells, covered and incubated for 1 hour at room temperature. The contents of the wells were decanted, and the wells were washed four times with wash solution. Next, 100 μL of Biotin-Antibody Conjugate was added to each well and the plate was covered and incubated at room temperature in the dark for 20 minutes. After incubation, the afore mentioned washing step was repeated and then 100 μL of HRP-Antibody Conjugate was added to each well. The plate was covered and incubated at room temperature in the dark for 20 minutes. The washing step was repeated. Next, 100 μL of Chromogen Substrate Solution

was added to each well and the plate was covered and incubated at room temperature in the dark for 10 minutes. Finally, 100μl of Stop Solution was added to each well. Plates were read at an absorbance of 450nm (BioTek Synergy H1 plate reader, BioTek Instruments, Winooski, VT) within 10 minutes of stop solution being added. The standard curve was linearized and fitted to a 5-parameter logistic curve using statistical software (GraphPad Prism). All samples were run in duplicate.

## Statistical analysis

Descriptive statistics for the patient populations were performed using Microsoft Excel for Mac (v. 16.16.27, 2016). Custom scripts using R (v4.2.1) were used to perform statical analyses (correlation, ANOVA) and to generate the plots (ROC curves, Kaplan-Meier). Multiple comparisons (stage of disease, clinical response, immunophenotype and substage) were performed using an ordinary one-way ANOVA with a Tukey HSD multiple pairwise comparisons test. Survival analysis was performed using a Kaplan-Meier survival curve. ROC curves were generated using the pROC R package v1.18.0. Confidence intervals were set at 95% using the Wilson/Brown method. The ROC curves AUC and the ANOVA p-values were both obtained from the 3 biomarkers levels based on the 3 defined disease states. Correlation coefficients were calculated using the R base corresponding function.

# Results

## Demographics and characteristics of the study population

The study population was composed of 40 dogs, 37 of which were diagnosed with lymphoma (LSA), two with acute myelogenous leukemia (AML) and one with multiple myeloma (MM). Thirty-two of the dogs with LSA were characterized as having multicentric disease, three had high grade cutaneous epitheliotropic and two had primary gastrointestinal lymphoma. Immunophenotype was available in 36 of the dogs with LSA. Twenty-four dogs (66%) had B cell immunophenotype, and 12 (33%) had T cell. Two of the T cells dogs were characterized as T-zone based on FLOW cytometry. Immunophenotyping was non-diagnostic in one dog due to low viability of the sample.

There were 18 spayed females, 19 neutered males, two intact males and one intact female in this cohort. The median age of enrolled dogs at diagnosis was 7 years (range 2–15, mean 7.8) and the median body weight was 29.74 kg (range 4.9–78, mean 28.4). A variety of breeds were enrolled in this study. Breeds included, mixed breed dog (n = 7), Labrador Retriever (n = 5), Pitt Bull Terrier (n = 3), Golden Retrievers (n = 3), English Mastiff (n = 3), Rottweiler (n = 2), Jack Russel Terrier (n = 2), German Shepard Dog (n = 2), and one each of the following breeds: Australian Cattle Dog, Australian Shepard, Boykin Spaniel, Boxer, Chow Chow, German Shorthaired Pointer, Leonberger, Maltese, Norwich Terrier, and Shih Tzu.

## Treatment

All dogs with any form of LSA were considered together unless otherwise specified. The two dogs with indolent lymphoma were not included in the survival analysis. Thirty-six dogs were enrolled at initial diagnosis and 4 dogs were enrolled at first relapse. Two of the dogs with relapsed LSA had multicentric B cell LSA (stages IVa and IVb) and both were treated with a second round of CHOP chemotherapy during this study. The other two dogs with relapsed LSA both had stage Va epitheliotropic LSA previously treated with chemotherapy. They were both rescued with radiation therapy while enrolled in this study. The remaining dogs were naïve to treatment at enrollment. Of the dog diagnosed with LSA, 31 were induced with

CHOP or L-CHOP. The remaining two dogs were induced with single agent doxorubicin (n = 1) or chlorambucil (n = 1). Both dogs with AML were treated with CHOP chemotherapy and the dog with MM received melphalan and prednisone.

## Outcomes

Thirty-three of the 37 dogs with LSA had documented progressive disease while on study. Both dogs with AML and the dog with MM also had documented disease progression while on study. At the end of the study period, 11 dogs were still alive and either in remission or undergoing treatment, 24 were dead or euthanized and five had been lost to follow up. Lymphoma was the cause of death in 21 of the dogs that died. For the remaining three, one developed hemophagocytic histiocytic sarcoma that was not responsive to treatment and necropsy confirmed no evidence of lymphoma at the time of death. One dog developed a lytic bone lesion, and the third dog developed seizure activity that progressed despite oral prednisone. While lymphoma recurrence cannot be completely ruled out in these cases both dogs originally presented with peripheral lymphadenopathy which was not present at the time of death.

Not surprisingly, there was a significant progression free survival (PFS) advantage (p<0.0001) for those dogs diagnosed with B cell lymphoma compared to those with non-indolent T cell lymphoma (PFS 305 days vs 114 days, respectively) (Table 1). The PFS for the two dogs with indolent T cell lymphoma was 210 and 138 days.

## Plasma H3.1-nucleosome concentrations at enrollment

Plasma H3.1-nucleosome concentrations were categorized as either high (>67.4 ng/mL), gray zone (>30.4 and ≤67.4 ng/mL) or low range (≤30.4 ng/mL) based on a standardized reference range for frozen plasma samples utilized by the Nu.Q Vet Cancer Test® (Fig 1). The median number of timepoints collected for each dog was 19 (range 5–85, mean 22.5). Thirty-one of the 40 dogs had elevated H3.1 nucleosome concentrations at enrollment (Fig 2). Among them, 28 of the 37 dogs (75.6%) with lymphoma demonstrated elevated H3.1 nucleosome levels at enrollment (Fig 2). Eighteen of these dogs had high levels of circulating plasma nucleosomes at enrollment (16 naïve, 2 relapsed) and 10 dogs were in the gray zone (8 naïve and 2 relapsed). Of these cases, five of the dogs with high levels and four of the dogs with levels in the gray zone were on prednisone at the time of initial sampling. Nine dogs had H3.1 plasma levels in the low range at diagnosis. Five of these dogs were on prednisone at the time of initial sampling. Both dogs with AML and the one dog with MM had high plasma nucleosome levels at diagnosis.

The median initial plasma H3.1 nucleosome concentration for all dogs was 70.1 ng/mL (mean 185.7, range 4.0–842.2, SD 248.1). The upper limit of detection for the assay is 842.2 ng/mL so all values above that (n = 3) were capped at 842.2 ng/mL for the purposes of analysis.

**Table 1. Characterization and outcomes of lymphoma cases based on immunophenotype.**

| Category | Median H3.1 concentration at enrollment (ng/mL) | Median PFS (days) | Median OST (days) |
|---|---|---|---|
| B cell (n = 24) | 56.52 (10.4–842.2) | 277 (21–933) | 490 (45–1026) |
| T cell (n = 10) | 84.02 (15–397.3) | 114* (22–248) | 232.5 (141–424) |
| T zone (n = 2) | 34.46 (4.02–64.9) | 210, 138 | 820, 215 |
| Unk (= 1) | 253.39 | 19 | 82 |

*Dogs with non-indolent T cell lymphoma had a significantly shorter progression free survival when compared to dogs with B cell lymphoma (p<0.0001). OST- overall survival time, LTF- Lost to follow up.

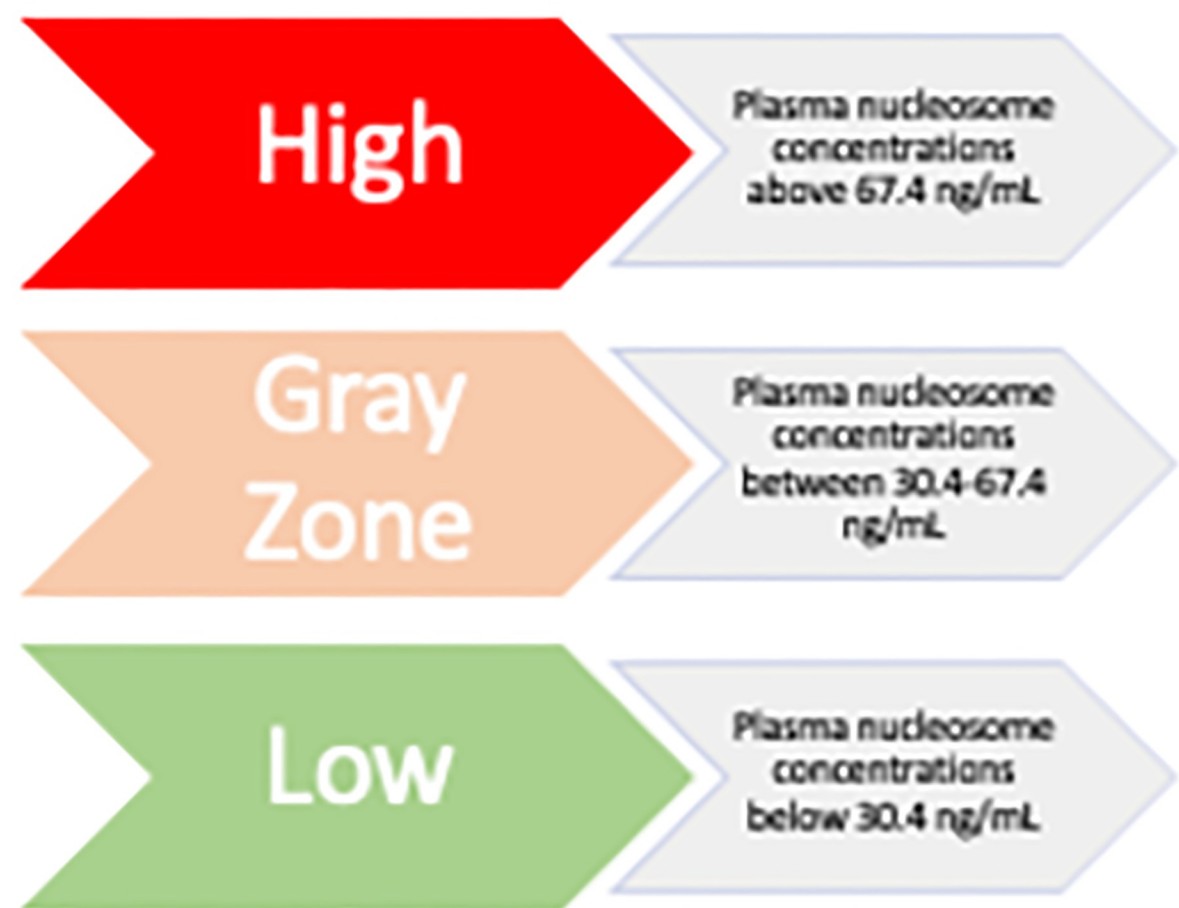

Figure 1: Detection reference range for the Nu.Q® Vet Cancer Test.

**Fig 1. Detection reference range for the Nu.Q® vet cancer test.**

The median plasma H3.1 concentration for dogs in the high range was 179.8 ng/mL (n = 20, mean 320.8, range 68.5–842.2, SD 281.7) and 53.0 ng/mL (n = 11, mean 53.1, range 30.6–64.9, SD 10.4) for dogs in the gray zone. Dogs with plasma H3.1 nucleosome levels in the low range had a median concentration of 15 ng/mL (n = 9, mean 17.91, range 4.0–29.1, SD 8.9) (Fig 2).

There was no significant difference between the median plasma H3.1 nucleosome concentrations at diagnosis based on immunophenotype (p = 0.66), stage (p≥0.98) or substage (p = 0.96) (Tables 1 and 2). However, there is a general increase in the median plasma

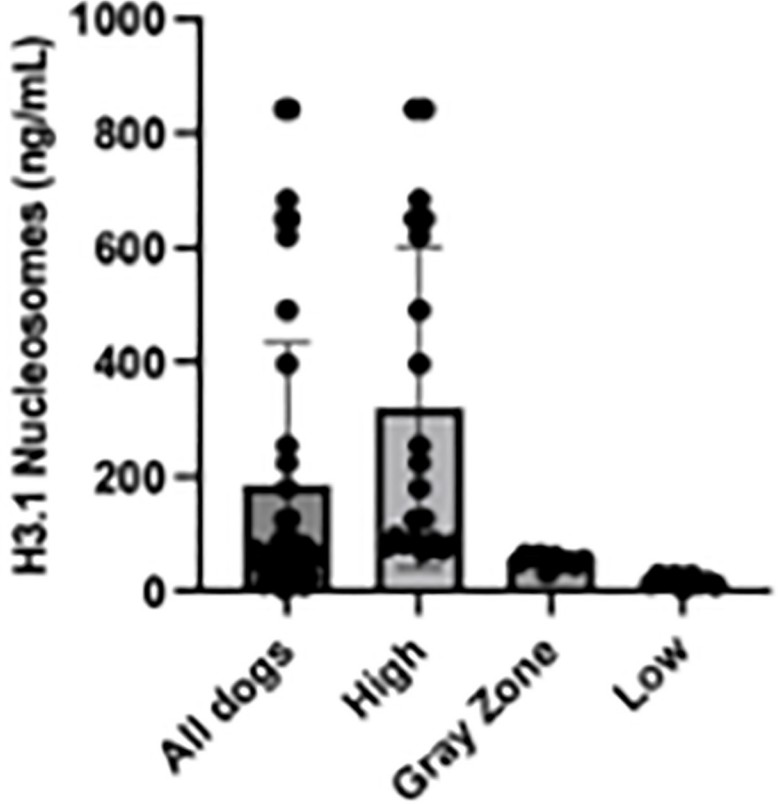

Figure 2: Plasma H3.1 nucleosome levels at diagnosis based on diagnostic categories in all dogs. Data is plotted as mean + standard deviation.

**Fig 2. Plasma H3.1 nucleosome levels at enrollment based on diagnostic categories in all dogs.** Data is plotted as mean + standard deviation.

nucleosome concentration as stage increases in these dogs which is consistent with previous reports [2]. The median plasma H3.1 nucleosome concentration for dogs with B cell lymphoma was 56.5 ng/mL (mean 161.5, range 10.4–842.2, SD 257.9) and the median for dogs with T cell lymphoma was 84.02 ng/mL (mean 124.4, range 15–397.3, SD 116.9). The plasma

Table 2. Characterization and outcomes of lymphoma cases based on stage (T-zone cases not included).

| Category | Median H3.1 concentration at enrollment (ng/mL) | Median PFS (days) | Median OST (days) |
| --- | --- | --- | --- |
| Stage III (n = 7) | 49.9 (10.4–842.2) | 239 (21–933) | 255 (45–933) |
| Stage IV (n = 19) | 60.9 (12.9–842.2) | 277 (64–844) | 518 (180–1026) |
| Stage V (n = 9) | 86.1 (22.3–491.4) | 139* (19–248) | 243 (82–445) |
| Substage a (n = 27) | 64.8 (10.4–842.2) | 254 (19–933) | 445 (45–1026) |
| Substage b (n = 8) | 65.2 (22.3–842.2) | 171** (71–262) | 381 (141–634) |

* p = 0.02 –stage V cases had a significantly shorter progression free survival regardless of phenotype when compared to stages III and IV.

** p = 0.05 –substage b cases has a significantly shorter progression free survival regardless of phenotype when compared to substage a.

H3.1 nucleosome concentrations for the two dogs with indolent T cell lymphoma were 64.9 and 4.015 ng/mL respectively.

## Plasma H3.1-nucleosome concentrations during the course of treatment showed lower levels at best response

Thirty-eight of the 40 dogs enrolled in this study had plasma H3.1 nucleosome concentrations remain or return to the low range during treatment. Nine dogs were in the low range at enrollment and 6 of these dogs remained in the low range throughout treatment. The median time to return to the low range (for those dogs with elevated plasma H3.1 nucleosome concentrations) was 17 days after initiating treatment (n = 29, mean 32.7, range 6–179 days, SD 39.05).

For the purposes of this study, the best clinical response was defined using RECIST criteria to determine a complete response (CR), a partial response (PR) or stable disease (SD). Progressive disease was also defined using RECIST criteria. Overall, 31 dogs achieved a CR (29 LSA, 1 AML and 1 MM), 7 dogs achieved a PR (6 LSA and 1 AML) and 2 achieved a SD (both LSA) as their best clinical response during the study period. The median time to best clinical response (documented in the medical record) was 22 days (n = 39, mean 28.49 days, range 6–86 days, SD 19.96). The two dogs whose plasma H3.1 nucleosome levels did not return to the low range during the study period had some of the shortest PFS times (21 and 26 days) and the shortest OS times (45 and 82 days) of all the dogs in this cohort. The median plasma H3.1 nucleosome concentration at enrollment for all dogs was significantly higher than the median at best response (p = 0.0016), but not significantly different from the concentration at progressive disease (p = 0.38). The median plasma H3.1 nucleosome concentration at the time of clinical best response for all dogs was 22 ng/mL (mean 28.5 ng/mL, range 3.8–128.6 ng/mL, SD 24.92) and this was significantly lower than the median plasma H3.1 nucleosome concentration at the time of clinical disease progression for all dogs (p = 0.0007) which was 61.13 ng/mL (mean 227.3 ng/mL, 6.65–973.0 ng/mL, SD 334) (Fig 3). The median plasma nucleosome concentration for those dogs achieving a CR (n = 31) was 22.6 ng/mL (range 3.8–55.2 ng/mL, mean 25.7, SD 14.6). The median plasma nucleosome concentration for those dogs whose best response was a PR (n = 7) was 21.8 ng/mL (range 4.13–68.2 ng/mL, mean 25.1, SD 21.9) and the median plasma nucleosome concentration for dogs whose best response was stable disease (n = 2) was 22.6 (range 10.7–34.5 ng/mL, mean 22.6, SD 16.8).

There was no significant difference in the time to best response for any group (median for dogs in the high zone 30 days, gray zone 18.5 days and low zone 17 days) (p>0.99). There was also no significant difference in the time it took dogs with elevated plasma H3.1 nucleosome concentrations to return to the low range (median for dogs in the high zone 14.5 days and gray zone 21 days) (p>0.99) when comparing all groups to each other. Finally, there was no significant difference in H3.1 plasma concentrations between dogs achieving a CR, PR or SD (p>0.86).

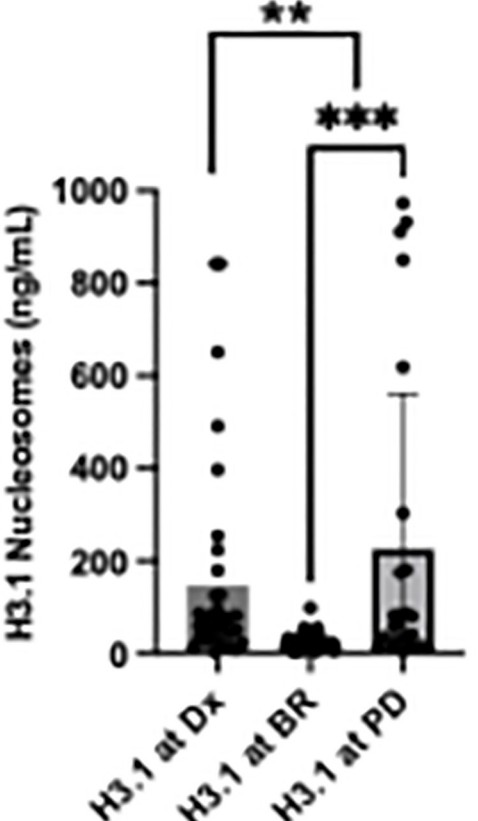

Fig 3. Comparison of plasma H3.1 concentrations at enrollment (Dx), progressive disease (PD) and best response for all dogs (BR).

However, for dogs with non-indolent LSA, a significant difference in PFS was noted with high plasma H3.1 nucleosome concentrations (>67.4 ng/mL; n = 18) when compared to those with gray zone or low plasma concentrations (<67.4 ng.ml; n = 17). Dogs with a high plasma H3.1 nucleosome concentration at enrollment had a median PFS of 219 days compared to those below 67.4 ng/mL with a median PFS of 305 days (p = 0.04) (Fig 4).

Twenty-six of 31 dogs (83.8%) with non-indolent LSA and documented disease progression had elevated plasma H3.1 nucleosome concentrations at the time of PD. These 31 dogs had 50

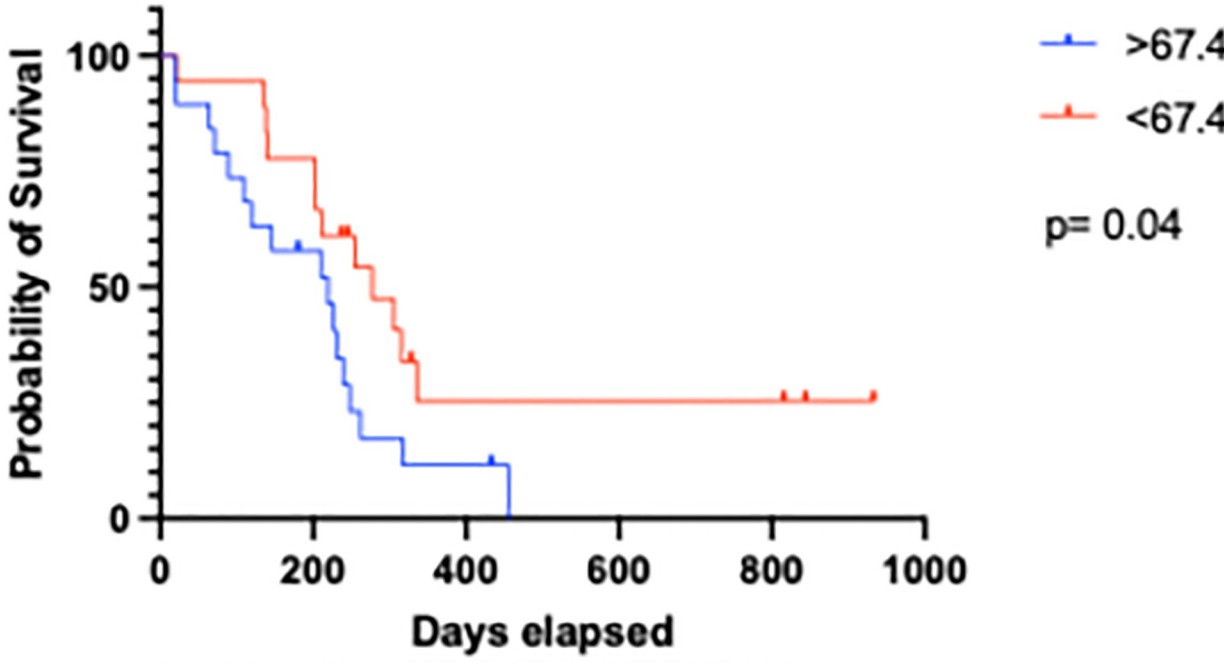

Figure 4. Kaplan Meyer curve demonstrating the difference in progression free survival for those cases with high plasma H3.1 nucleosome levels at diagnosis versus those with low or gray zone H3.1 plasma concentrations at diagnosis.

**Fig 4. Kaplan Meyer curve demonstrating the difference in progression free survival for non-indolent lymphoma cases with high plasma H3.1 nucleosome levels at enrollment versus those with low or gray zone H3.1 plasma concentrations at enrollment.**

clinically determined PD events altogether. Of these, 35 events had elevated plasma H3.1 nucleosome levels and 15 were in the low range. Three dogs were noted to have plasma H3.1 nucleosome levels in the low range at enrollment (all three were on prednisone at initial sample collection) but had elevated levels at 7 of their 8 combined PD events. There were 6 dogs that originally had elevated plasma H3.1 nucleosome levels that were in the low range when PD was noted for 10 of their combined 16 PD events.

Thirteen dogs had elevated plasma H3.1 nucleosome concentrations before progressive disease was noted. These 13 dogs had 19 progressive events altogether. The median time from plasma H3.1 nucleosome concentration elevation to clinical documentation of PD was 10.5 days (mean 18.15 days, range 6–80 days).

## Correlation of CRP and TK1 to H3.1 plasma concentrations over time

There was no correlation between plasma H3.1 nucleosome and CRP or TK1 at any of these time points (Fig 5). Furthermore, additional interrogation of plasma H3.1 nucleosome, CRP

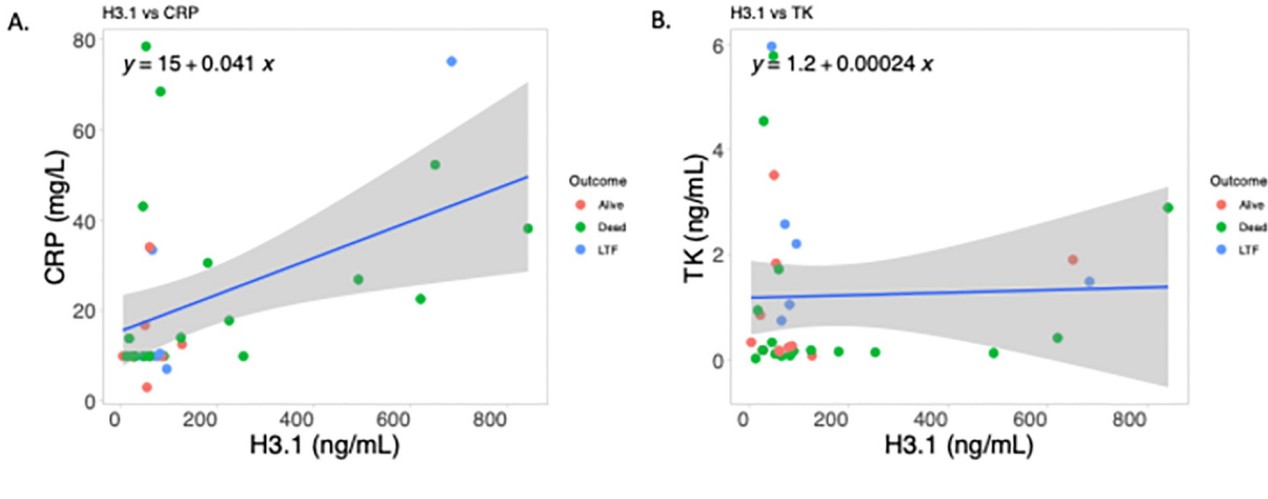

Figure 5: These scatter plots compare the plasma concentrations for plasma H3.1 nucleosome versus CRP (A.) and TK1 (B.) for each dog based on outcome at the end of the study period. (LTF- lost to follow up). Neither CRP nor TK1 correlated well with plasma H3.1 nucleosome concentrations during the study.

**Fig 5.** These scatter plots compare the plasma concentrations for plasma H3.1 nucleosome versus CRP (A.) and TK1 (B.) for each dog based on outcome at the end of the study period. (LTF- lost to follow up). Neither CRP nor TK1 correlated well with plasma H3.1 nucleosome concentrations during the study.

and TK1 as biomarkers for treatment and remission monitoring were performed in this cohort. Plasma H3.1 nucleosome concentrations correctly differentiate dogs with active disease (enrollment or progressive disease), moderate disease (stable disease or a partial remission) and clinical remission at a significantly higher rate than the other two biomarkers (Table 3) (Fig 6). There was an increase in TK1 in dogs in remission such that TK1 was significantly higher in dogs in remission than dogs with active disease and dogs with moderate disease (Table 3).

## Unexplained elevations in H3.1 plasma concentrations during treatment

During the treatment period 18 dogs (45%) experienced 31 elevations in plasma H3.1 nucleosome concentrations that could not be explained by disease progression out of 803 total sample collections (3.8%). These cases can be divided into 3 groups. The first group are dogs with elevated neutrophil counts and known concomitant inflammatory conditions. Eight (25.8%) dogs fall into this group and had conditions such as a diffuse deep skin infection in a dog with epitheliotropic lymphoma, cellulitis requiring hospitalization, pneumonia and a dog fight. CRP was also elevated in four of these cases. The second group includes fourteen (45.2%) dogs with normal neutrophil counts and no obvious concomitant diseases. CRP was also elevated in four of these cases and TK1 was elevated in two of these cases. One of the dogs that had a

**Table 3. Mean concentrations of plasma H3.1 nucleosome, CRP and TK1 levels based on disease state.**

| Disease state | Mean H3.1 (ng/mL) | Mean CRP (mg/L) | Mean TK1 (ng/mL) |
|---|---|---|---|
| Active (Dx and PD) | 171.26 | 23.30 | 0.74 |
| Moderate (SD and PR) | 23.85 | 12.99 | 0.61 |
| Remission (CR) | 38.22 | 12.51 | 1.36 |

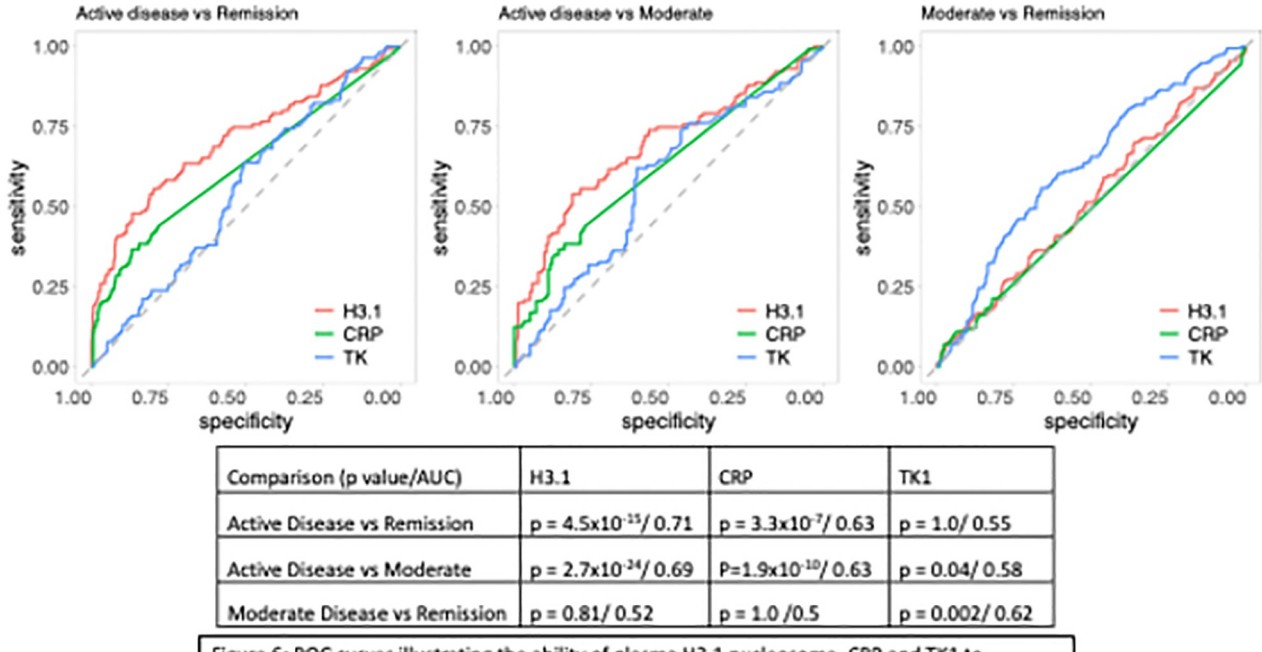

| Comparison (p value/AUC) | H3.1 | CRP | TK1 |
|---|---|---|---|
| Active Disease vs Remission | p = 4.5x10^-15 / 0.71 | p = 3.3x10^-7 / 0.63 | p = 1.0 / 0.55 |
| Active Disease vs Moderate | p = 2.7x10^-24 / 0.69 | P=1.9x10^-10 / 0.63 | p = 0.04 / 0.58 |
| Moderate Disease vs Remission | p = 0.81 / 0.52 | p = 1.0 / 0.5 | p = 0.002 / 0.62 |

Figure 6: ROC curves illustrating the ability of plasma H3.1 nucleosome, CRP and TK1 to differentiate between active disease (at diagnosis or progressive disease), moderate disease (stable disease or partial remission) and remission (clinical remission). Area under the curve and p values are reported for each graph in the table under the graphs.

**Fig 6. ROC curves illustrating the ability of plasma H3.1 nucleosome, CRP and TK1 to differentiate between active disease (at enrollment or progressive disease), moderate disease (stable disease or partial remission) and remission (clinical remission).** Area under the curve and p values are reported for each graph in the table under the graphs.

normal neutrophil count and elevated CRP also had a toe infection severe enough to warrant a treatment delay. The last group consisted of five (16.1%) dogs with neutrophil counts below the reference range requiring treatments delays as a result. In these cases, neutrophil attrition secondary to chemotherapy along with the acute phase proteins enacted to restore normal neutrophil counts could explain these elevations [29]. There were three elevations (9.6%) that were noted at times when the dogs did not have concomitant diseases, were in remission and did not have CBCs performed during the visit.

## Discussion

Previously published studies demonstrated frequent elevations in plasma nucleosome concentrations in dogs with lymphoma at diagnosis [2, 27], however, those were single time point studies and did not evaluate the utility of monitoring nucleosomes throughout treatment. For the first time we demonstrate that cfDNA levels, in the form of nucleosomes can be used to monitor cancer progression and remission. In the current study, we show that nucleosome levels are not only elevated at diagnosis, but they nearly always return to the low range during treatment and are associated with clinical best response suggesting that the disease burden drops below this threshold and can be cleared from the body at a normal rate. Furthermore, nucleosome elevations often recur at the time of disease progression, mirroring the clinical course of the disease and that plasma nucleosome concentrations above 67.4 ng/mL at diagnosis are inversely correlated with survival. A study presented at the ISTH 2021 virtual conference demonstrated rapid rises and decreases in serial measurements of H3.1 nucleosome levels

within a 24-hour period in human patients with COVID19. This data indicates that the half-life of nucleosomes in the circulation must be sufficiently long, in addition to an increase in rate of release. Similarly, detection of a rapid fall in nucleosome levels would not be possible if the half-life were very long in relation to the change in the release rate. Therefore, the half-life of nucleosomes is sufficiently long to detect rapid rises, but not so long as to prevent detection of rapid decreases [30]. Taken together these results demonstrate that monitoring nucleosome levels can be an important tool in monitoring disease and treatment response in both dogs and humans, enabling clinicians to more optimally deliver treatment regimes.

In the current study, nucleosome levels were not statistically different based on stage, however, there were no dogs enrolled in this study at a stage lower than III. In humans, plasma nucleosome concentrations were able to correctly differentiate disease stage and the presence of metastatic disease only in patients with gastrointestinal cancer (including colorectal cancer) [24, 25]. This type of disease stratification has not been found in other types of human cancers studied including lung cancer, breast cancer, ovarian and other gynecologic cancers, renal and prostatic cancer as well as lymphoma [31]. Dogs with B cell lymphomas did have a higher mean plasma nucleosome concentration, as has been previously reported, but this was not statistically significant [2]. While dogs with substage a disease had lower plasma nucleosome concentrations at enrollment, this was also not significant.

Many of the dogs (n = 14) enrolled in this study were started on prednisone before presentation to the oncology service preventing the collection of a truly naïve baseline sample. Oral prednisone therapy often results in remission durations of 1–2 months [32]. The fact that many dogs were on prednisone at the time of enrollment could have falsely decreased their initial H3.1 plasma concentration levels due to a response to this steroid. Three of the dogs that initially presented with plasma H3.1 concentrations in the low range while on prednisone developed elevated plasma nucleosome concentrations at the time of progression. Caution is recommended when interpreting initial plasma H3.1 nucleosome concentrations for patients with hematopoietic malignancies while on prednisone. However, in the face of chemotherapy resistance, progressive disease did result in elevated nucleosome concentrations in the majority of cases in this cohort.

Elevated plasma nucleosome concentrations have been linked to tumor burden and disease progression for multiple malignancies in humans even in early disease stages [25, 33, 34]. In this study, dogs with high plasma H3.1 nucleosome concentrations at enrollment had a significantly shorter median PFS when compared to dogs with lower concentrations. For these cases, H3.1 plasma nucleosome levels may represent a surrogate of total disease burden unrelated to disease stage. In humans, there is some support for the use of plasma nucleosome concentrations for prognostication in patients with advanced lung cancer, but its use as a prognostic tool has not held true for other types of malignancies [25]. Additionally, plasma H3.1 nucleosome concentrations were able to predict disease state more reliably than TK1 and to a lesser degree, CRP plasma levels.

Plasma nucleosome concentrations are not specific for neoplastic disease. Any disease associated with accelerated cell death, such as autoimmune disease, trauma or severe infections, can lead to increases in plasma nucleosome concentrations [24]. In this study, several dogs had unexplained elevations in plasma H3.1 concentrations that were unrelated to clinical cancer observation during their treatments. These could have been caused by a failure to appropriately fast patients before blood draws, as not fasting could causes plasma lipids or other metabolites to interfere with the test, sample mishandling or an unidentified nidus of inflammation. Unexplained spikes in the neutrophil counts accompanied many of these unexplained elevations, indicating that the elevated nucleosome levels could be due to ongoing NETosis. Additionally, elevations of plasma H3.1 nucleosome concentrations were also seen when

concurrent diseases, such as pneumonia or cellulitis, were present. When used as a monitoring tool, care should be taken to ensure that no other concurrent diseases could be causing an increased plasma H3.1 nucleosome concentration, that the patient is fasted and the sample is handled appropriately. Furthermore, approximately 25% of dogs with lymphoma will not have elevated nucleosome concentrations at diagnosis. Based on the results of this study, plasma nucleosome concentrations are likely not useful for disease monitoring in dogs with lymphoma that do not have an elevated nucleosome concentration at diagnosis and are not currently on prednisone.

When used as a monitoring tool, care should It is recommended that at least two consecutive elevations several days apart be documented without evidence of other underlying conditions before consideration should be given to disease progression. In this study, none of the aberrantly elevated nucleosome concentrations were elevated for 2 consecutive weeks without a known concomitant inflammatory cause. Due to the nonspecific nature of plasma nucleosomes, additional confirmatory tests will be needed to determine progressive disease. However, these levels can act as a signal of disease progression warranting more frequent monitoring or additional tests, such as lymph node cytology, which otherwise would not have been performed.

In this study, elevated plasma nucleosome concentrations preceded clinical detection of progression 19 times in 13 dogs. Plasma nucleosome concentrations were elevated a median of 10.5 days before clinical detection occurred. Furthermore, nucleosome concentrations can change dramatically from week to week as patients respond to chemotherapy, as is seen in S1–S3 Figs. These cases demonstrate the dynamic nature of plasma nucleosome concentrations in 3 dogs with relapsed LSA, AML and naïve LSA, respectively. Of note, in the first supplemental case, the plasma nucleosome concentration did not increase when a large benign splenic mass was identified during routine staging but did increase as the patient developed histiocytic sarcoma. In the 3rd case, plasma H3.1 concentrations were significantly better at predicting disease progression than CRP or TK1. Many of the dogs in the study that developed progressive disease were noted to have increases in their plasma nucleosome concentrations at the time of progressive disease. However, more than half of these cases were noted to have elevated plasma nucleosome concentrations during treatment when they were being seen several times per month. Testing this often did not improve the test's ability to detect progressive disease. It is recommended that when using plasma nucleosome concentrations to monitor treatment response that H3.1 nucleosome concentrations be evaluated at the beginning of each cycle and at each follow up visit after chemotherapy has been discontinued.

## Conclusion

When applied in the correct context, plasma nucleosome concentrations can be a useful addition to the veterinary oncology toolbox for monitoring disease response and remission for cases with known hematopoietic malignancies.

## Supporting information

**S1 Fig. H3.1 trends for a 6 y/o FS Rottweiler presenting 9 months after finishing CHOP #1 out of remission.** CHOP was re-initiated and mitoxantrone was substituted for doxorubicin starting with cycle 2. She presented with 7 cm cavitated splenic mass 3 months after completing CHOP and underwent splenectomy. The histopathology diagnosed benign fibroplasia. Her values begin to rise at the 10th recheck and she was diagnosed with Histiocytic Sarcoma Hemophagic Syndrome after a bone marrow aspirate. She was treated with CCNU but did not respond well enough and was euthanized 2 weeks after diagnosis. The red circles represent

new diagnosis or progressive disease, and the green circles represent clinical remission as noted in the medical record. Del- treatment delay, Vinc- vincristine, Cytox- Cyclophosphamide, Doxo-doxorubicin, Mitox- Mitoxantrone.
(TIF)

**S2 Fig. H3.1 trends during treatment for a 7 y/o MN mixed breed dog with AML.** He presented initially on a high dose of prednisone for 1 week. He did initially respond to CHOP therapy, however, progressed quickly and was euthanized shortly after receiving L-spar.
(TIF)

**S3 Fig. H3.1 trends for a 3 y/o FS Australian Shepherd newly diagnosed with lymphoma.** Several samples unable to be collected as some cyclophosphamide doses were given at home and 2 vincristine doses were missed due to COVID19 disruptions in the clinic. H3.1 values start to trend up in the last cycle which ended with a high.
(TIF)

## Author Contributions

**Conceptualization:** Heather Wilson-Robles.

**Data curation:** Heather Wilson-Robles, Emma Warry, Tasha Miller, Jill Jarvis, Matthew Matsushita.

**Formal analysis:** Heather Wilson-Robles, Marielle Herzog, Jean-Valery Turatsinze.

**Funding acquisition:** Heather Wilson-Robles.

**Investigation:** Heather Wilson-Robles, Emma Warry, Jill Jarvis.

**Methodology:** Heather Wilson-Robles, Tasha Miller, Pamela Miller.

**Project administration:** Heather Wilson-Robles, Emma Warry, Jill Jarvis.

**Resources:** Heather Wilson-Robles, Emma Warry, Tasha Miller, Jill Jarvis, Matthew Matsushita, Pamela Miller, Marielle Herzog, Jean-Valery Turatsinze, Theresa K. Kelly, S. Thomas Butera, Gaetan Michel.

**Software:** Heather Wilson-Robles, Marielle Herzog, Jean-Valery Turatsinze.

**Supervision:** Heather Wilson-Robles.

**Validation:** Heather Wilson-Robles, Tasha Miller.

**Visualization:** Heather Wilson-Robles.

**Writing – original draft:** Heather Wilson-Robles, Emma Warry, Marielle Herzog, Jean-Valery Turatsinze, S. Thomas Butera.

**Writing – review & editing:** Heather Wilson-Robles, Emma Warry, Jill Jarvis, Matthew Matsushita, Pamela Miller, Marielle Herzog, Jean-Valery Turatsinze, Theresa K. Kelly, S. Thomas Butera, Gaetan Michel.

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
