## [Decision Letter · Decision Letter 0]

1 Mar 2023

PONE-D-23-02803Monitoring plasma nucleosome concentrations to measure of disease response and progression in dogs with hematopoietic malignancies.PLOS ONE

Dear Dr. Wilson-Robles,

Thank you for submitting your manuscript to PLOS ONE. After careful consideration, we feel that it has merit but does not fully meet PLOS ONE’s publication criteria as it currently stands. Therefore, we invite you to submit a revised version of the manuscript that addresses the points raised during the review process.

Please address all Reviewer comments.  I agree with the Reviewers that inclusion of multiple individual case vignettes is inappropriate in the manuscript body. If you would like to include these, or a subset, they can be included as Supplemental Data.

We look forward to receiving your revised manuscript.

Kind regards,

Douglas H. Thamm, V.M.D.

Academic Editor

PLOS ONE

Funding for materials and author salaries was provided by Belgian Volition SPRL. HWR and TM received salary from these funds. JVT, TK, MH, TB and GM are employees of Belgian. Volition & Volition America. The URL to the Belgian Volition website is: https://volition.com/. Some additional funding for HWR and TM salaries was provided by the Fred and Vola Palmer Chair of Comparative Oncology held by HWR.”

Reviewers' comments:

Reviewer's Responses to Questions

**Comments to the Author**

1. Is the manuscript technically sound, and do the data support the conclusions?

Reviewer #1: Yes

Reviewer #2: Yes

2. Has the statistical analysis been performed appropriately and rigorously? 

Reviewer #1: Yes

Reviewer #2: Yes

3. Have the authors made all data underlying the findings in their manuscript fully available?

Reviewer #1: Yes

Reviewer #2: Yes

4. Is the manuscript presented in an intelligible fashion and written in standard English?

Reviewer #1: Yes

Reviewer #2: Yes

5. Review Comments to the Author

Reviewer #1: Thank you for the opportunity to review this important work. The study was well thought out and the manuscript well written. I do have a few comments/edits as described below:

Line 77: typo – remove one of the ‘ongoings’ in this sentence

Line 90/91: “Plasma nucleosome levels increase when the concentration within the blood exceeds the body’s ability to clear them or when they bind to acute phase proteins.” Can you comment on whether it is known if renal impairment reduces the rate of nucleosome clearance?

Line 189 and Line 201: Please comment on your reason for including T zone dogs in this study. Inclusion of T zone dogs is interesting as you well know this disease has indolent characteristics and thus differs significantly from the standard large cell lymphoma case. You mentioned in line 201 that all the LSA cases were grouped into one – how could including the indolent cases impact the outcome of the data you present?

Line 294-298: Please clarify the number of dogs in each group (ie the high concentration group vs the gray zone/low concentration group). I had a difficult time sorting this out in the text of the manuscript – please clarify here. “However, for dogs with LSA, a significant difference in PFS was noted with high plasma H3.1 295 nucleosome concentrations (>67.4 ng/mL) when compared to those with gray zone or low 296 plasma concentrations.”

Not surprisingly, stage 5 and substage b dogs had a poorer outcome; however, interestingly their nucleosome concentration prior to starting treatment was not much different compared to the other groups – please comment on this in the context of why this could be as well as how this testing will help with outcome assessment in dogs with this stage/substage.

Line 492-494: “It is recommended that when using plasma nucleosome concentrations to monitor treatment response that H3.1 nucleosome concentrations be evaluated at the beginning of each cycle and at each follow up visit after chemotherapy has been discontinued.” Please comment on how this testing will alter a clinicians treatment recommendation (i.e. will a clinician recommend restarting treatment with elevation of nucleosome levels, in the absence of other clinical indicators of PD – the concern here is that this elevation is not specific to cancer and can be impacted by inflammation as you previously stated)

Reviewer #2: This manuscript reports the results of a prospective clinical trials assessing the role of nucleosome measurements in remission monitoring in dogs with hematologic malignancies. The study is well conducted and results have significant value. The inclusion of 3 dogs with diagnoses other than LSA was perhaps not ideal and adds minimally (if at all) to the value of this manuscript. Overall, the manuscript is well written, but there are several areas that require clarification (listed below). The individual case reports are interesting, but may be more appropriate to report as supplemental data. Additionally, the discussion must more fully discuss the possibility of false negative results and the potential reasons for and clinical impacts of those results.

Introduction:

Line 77: Change wording to not use “ongoing” twice.

Line 82-83: Clarify whether you mean that neutrophils are the “specialized immune cell” or if you mean a different cell type

Line 84-85: Clarify briefly what role nucleosomes play in cellular functions other than providing DNA scaffolding mentioned above

Methods:

Line 115: Were RECIST criteria utilized to determine disease status?

Line 127: Specify the timepoints at which samples were collected. Predetermined? Relapse? Clinical remission?

Line 152: Specify how results of the nucleosome assay were interpreted.

Line 154: Specify type of sample (plasma?)

Results:

Line 187: Clarify whether you mean cutaneous epitheliotropic (and correct misspelling) LSA.

Please also specify grade of disease for multicentric and GI LSA cases

Lines 201-210: This section describing therapy is confusing to the reader. Clarify which therapies were administered during the study period. Previous therapies are less important and can be removed or described in less detail.

Line 216: Change “hemophagic” to “hemophagocytic”

Lines 216-219: This long sentence is hard to read. Consider making 2 sentences.

Line 231: Change “40 samples” to “40 dogs” and change “diagnosis” to “enrollment” since some cases were relapsed. Make this change throughout the manuscript.

Line 233: Change “which” to “these dogs”

Line 236-237: This statement that steroids may falsely decrease H3.1 plasma levels is new information and affects the reader’s interpretation of the data significantly. Consider just reporting numbers of dogs on steroids here and then include information about why and to what degree this may occur in discussion and/or intro.

Line 281: The readers need more information about exactly what “best response” was in these dogs and how it correlated with H3.1 concentration. For example, how many dogs achieved CR, PR, SD, and PD and how did their H3.1 concentrations differ from dogs with SD or PD?

Lines 289-293: Please be more clear as to what exact statistical comparison you are reporting here and throughout the manuscript. For example, “There was no significant difference in time to best response based on H3.1 nucleosome results” or similar.

Lines 316-320: Most of this description of methodology should be in the methods section of the manuscript.

Line 335: Consider listing total number of samples collected at beginning of results section along with total number of dogs.

Line 351: Consider adding the word “obvious” between “no” and “concomitant diseases” since it is impossible to rule out undiagnosed conditions

Lines 352-359: It is difficult to tell which of these dogs were in which groups. Consider starting by dividing all of these collections into major categories (elevated neutrophils and concomitant conditions, normal neutrophils and concomitant conditions, normal neutrophils no known concomitant conditions). Then discuss subgroups as appropriate.

Case Examples: I will defer to the editor, but suggest that these case examples may be most appropriate in supplemental materials and not in the main text of the manuscript.

Line 366: It is not clear when this dog was actually enrolled in the study.

Line 401: Change “great response” to “complete remission” or similar

Discussion:

Line 420: Your statement about nucleosome levels being associated with clinical remission may be true, but readers have not seen that specific data. You show lower H3.1 levels at “best response” but do not define that response or associate those responses with H3.1. Please clearly report comparisons between remission status using RECIST criteria and H3.1 concentrations in the results.

Line 423: Clarify that higher nucleosome levels “before treatment” are inversely correlated with survival (or similar).

Lines 446-448: I assume that dogs in the trial had to have gross disease to enroll, so it seems that dogs out of remission or not in complete remission should still have elevated H3.1 levels. Are there other reasons steroids could affect H3.1 levels? A false negative results is also possible and should be discussed here.

Lines 462-464: This last sentence is confusing because of its placement after a discussion about human malignancies. State whether you are referring do your dog data or human data. If dog data, consider additional discussion interpreting this result or delete from this part of the discussion.

Line 470: More discussion about the role of fasting in accuracy of H3.1 testing is warranted.

Line 488: How often did you see H3.1 level increases preceeding relapse? How long before relapse was clinically evident? This information would be very helpful for clinical veterinarians to know.

Line 489: Do you mean one time elevated concentrations? Concentrations that returned to normal without intervention? Please clarify what you mean here.

Line 490-491: Because you did not report PPV, reword this sentence to be clear that this statement is a clinical interpretation of the data and not a statistical result.

Discussion general: The discussion adequately discusses the possibility of false positive H3.1 results, but does not adequately discuss the possibility of or causes of false negative results. Please include more discussion of this and how it could affect clinical interpretation. Additionally, there appears to be some inconsistency in the interpretation of gray zone results in the manuscript. Discussion of how to interpret a gray zone result is warranted.

Table 1:

Clarify footnote to state which group you compared the non-indolent T-cell LSA dogs PFS to.

Be consistent with reporting mean or median H3.1 concentrations in groups. If reporting medians, include range. If reporting means, include SD.

Adding reference range for H3.1 concentration in the legend for table 1 would aid readers in interpretation.

Clarify whether number labeled as PFS and OST are medians. Also report range.

The last column with censoring and individual case information does not add to the table and should be removed.

Table 2:

Be consistent with reporting mean or median H3.1 concentrations in groups. If reporting medians, include range. If reporting means, include SD.

Clarify whether number listed for PFS and OST is a median. Also report range.

The last column with censoring and individual case information does not add to the table and should be removed.

Only one significant digit need be included when reporting p-values.

Clarify footnotes to state which groups these cases were compared to.

Figure 2.

Specify that these data are from the time of study enrollment

Figure 3.

This figure would make more sense if best response data was between Diagnosis (enrollment) and PD since that would happen chronologically

Figure 4. Are these data just LSA dogs or all dogs? Please clarify in the legend.

Figures 7-9: Considering moving with text to supplemental materials.

6. PLOS authors have the option to publish the peer review history of their article (what does this mean?). If published, this will include your full peer review and any attached files.

Reviewer #1: No

Reviewer #2: No

---

## [Author Response · Author response to Decision Letter 0]

10 Mar 2023

Response to reviewers

The authors would like to thank the reviewer for their time and thoughtful consideration of our manuscript. We are grateful for the comments and have tried our best to effectively answer these comments and questions to your satisfaction. 

A section has been added to the methods section addressing these issues. 

Funding for materials and author salaries was provided by Belgian Volition SPRL. HWR and TM received salary from these funds. JVT, TK, MH, TB and GM are employees of Belgian. Volition & Volition America. The URL to the Belgian Volition website is: https://volition.com/. Some additional funding for HWR and TM salaries was provided by the Fred and Vola Palmer Chair of Comparative Oncology held by HWR.”

This statement has been amended as requested on the cover letter. 

This has been corrected as requested. 

This has been deleted from the manuscript after the references. 

Reviewers' comments:

Reviewer's Responses to Questions

The authors would like to thank the reviewer for their time and thoughtful consideration of our manuscript. We are grateful for the comments and have tried our best to effectively answer these comments and questions to your satisfaction. 

Comments to the Author

1. Is the manuscript technically sound, and do the data support the conclusions?

Reviewer #1: Yes

Reviewer #2: Yes

2. Has the statistical analysis been performed appropriately and rigorously? 

Reviewer #1: Yes

Reviewer #2: Yes

3. Have the authors made all data underlying the findings in their manuscript fully available?

Reviewer #1: Yes

Reviewer #2: Yes

4. Is the manuscript presented in an intelligible fashion and written in standard English?

Reviewer #1: Yes

Reviewer #2: Yes

5. Review Comments to the Author

Reviewer #1: Thank you for the opportunity to review this important work. The study was well thought out and the manuscript well written. I do have a few comments/edits as described below:

Line 77: typo – remove one of the ‘ongoings’ in this sentence

This has been corrected as requested. 

Line 90/91: “Plasma nucleosome levels increase when the concentration within the blood exceeds the body’s ability to clear them or when they bind to acute phase proteins.” Can you comment on whether it is known if renal impairment reduces the rate of nucleosome clearance?

 In humans, end stage renal disease and patients who are on hemodialysis for end stage renal disease do have elevated levels of plasma nucleosomes, though inflammation is thought to play a key role in these cases more so than reduced clearance. This has not been evaluated in dogs. We do have a handful of cases with severe kidney disease or acute on chronic kidney injury and many of them to do have elevated nucleosome concentrations, however, these dogs were all sick enough to present to the ER and this co-morbidity was easily recognized (none of the dogs in this study had any level of kidney insufficiency). Due to the lack of published information available for the dog, we did not include a statement regarding this information in this manuscript but I have included 2 references below related to end stage kidney disease in humans and circulating nucleosome concentrations. 

Additionally, a statement has been added to the methods sections stating that dogs could not have significant co-morbidities (lines 119-120).

-Elevated extracellular nucleosomes and their relevance to inflammation in stage 5 chronic kidney disease. Phan T, Mcmillan R, Skiadopoulos L, Walborn A, Hoppensteadt D, Fareed J, Bansal V. Int Angiol. 2018 Oct;37(5):419-426. doi: 10.23736/S0392-9590.18.03987-1. Epub 2018 Apr 11. PMID: 29644836

-Prognostic role of circulating neutrophil extracellular traps levels for long-term mortality in new end-stage renal disease patients. Kim JK, Lee HW, Joo N, Lee HS, Song YR, Kim HJ, Kim SG. Clin Immunol. 2020 Jan;210:108263. doi: 10.1016/j.clim.2019.108263. Epub 2019 Oct 17. PMID: 31629808

Line 189 and Line 201: Please comment on your reason for including T zone dogs in this study. Inclusion of T zone dogs is interesting as you well know this disease has indolent characteristics and thus differs significantly from the standard large cell lymphoma case. You mentioned in line 201 that all the LSA cases were grouped into one – how could including the indolent cases impact the outcome of the data you present?

 This has been clarified to state that non-indolent lymphomas were not included in the survival analysis. 

Line 294-298: Please clarify the number of dogs in each group (ie the high concentration group vs the gray zone/low concentration group). I had a difficult time sorting this out in the text of the manuscript – please clarify here. “However, for dogs with LSA, a significant difference in PFS was noted with high plasma H3.1 295 nucleosome concentrations (>67.4 ng/mL) when compared to those with gray zone or low 296 plasma concentrations.”

These case numbers have been added – now lines 362-363.

Not surprisingly, stage 5 and substage b dogs had a poorer outcome; however, interestingly their nucleosome concentration prior to starting treatment was not much different compared to the other groups – please comment on this in the context of why this could be as well as how this testing will help with outcome assessment in dogs with this stage/substage.

Plasma nucleosome concentrations do increase with increasing stage, though this is not clinically significant. We also did not have any dogs with stage I or II disease in this study, and therefore cannot say whether is a difference in the nucleosome concentration between a stage V and stage I dog. While it is true that cell free DNA often does increase with disease burden, the staging schemes are typically associated with location rather than disease burden, so it is possible (and indeed seen here) that some dogs with stage III or IV disease had an equal or higher disease burden than a dog with stage V who may have only had GI or skin involvement. The same holds true for substage, while dogs with substage b disease often do have a higher disease burden than those with substage a disease, this is not always true and may skew the results so that they are not clinically significant. Additionally, disease stratification with plasma nucleosome concentrations based on stage is not seen in many cancers including lung, breast ovarian, renal, prostatic and lymphoma in humans (lines 566-575 of discussion) and it appears this may be true in the dog as well.

Line 492-494: “It is recommended that when using plasma nucleosome concentrations to monitor treatment response that H3.1 nucleosome concentrations be evaluated at the beginning of each cycle and at each follow up visit after chemotherapy has been discontinued.” Please comment on how this testing will alter a clinicians treatment recommendation (i.e. will a clinician recommend restarting treatment with elevation of nucleosome levels, in the absence of other clinical indicators of PD – the concern here is that this elevation is not specific to cancer and can be impacted by inflammation as you previously stated)

We have tried to expand upon this comment in lines 652-660.

Reviewer #2: This manuscript reports the results of a prospective clinical trials assessing the role of nucleosome measurements in remission monitoring in dogs with hematologic malignancies. The study is well conducted and results have significant value. The inclusion of 3 dogs with diagnoses other than LSA was perhaps not ideal and adds minimally (if at all) to the value of this manuscript. Overall, the manuscript is well written, but there are several areas that require clarification (listed below). The individual case reports are interesting, but may be more appropriate to report as supplemental data. Additionally, the discussion must more fully discuss the possibility of false negative results and the potential reasons for and clinical impacts of those results.

Introduction:

Line 77: Change wording to not use “ongoing” twice.

 This has been changed as requested.

Line 82-83: Clarify whether you mean that neutrophils are the “specialized immune cell” or if you mean a different cell type

 This has been clarified.

Line 84-85: Clarify briefly what role nucleosomes play in cellular functions other than providing DNA scaffolding mentioned above

This has been clarified in lines 85-86.

Methods:

Line 115: Were RECIST criteria utilized to determine disease status?

 RECIST criteria were used and this has been added (line 122). 

Line 127: Specify the timepoints at which samples were collected. Predetermined? Relapse? Clinical remission?

 Timepoints for collection have been added here (lines 135-137).

Line 152: Specify how results of the nucleosome assay were interpreted.

 An expanded description of the algorithm has been added here (lines 159-164).

Line 154: Specify type of sample (plasma?)

The word plasma does precede the first written description for C-reactive protein. Please let us know if additional clarification is needed. 

Results:

Line 187: Clarify whether you mean cutaneous epitheliotropic (and correct misspelling) LSA.

Please also specify grade of disease for multicentric and GI LSA cases

This clarification has been made as requested (line 206).

Lines 201-210: This section describing therapy is confusing to the reader. Clarify which therapies were administered during the study period. Previous therapies are less important and can be removed or described in less detail.

We have attempted to clarify the paragraph so that it is, hopefully more streamlined and easier to understand (lines 223-231)

Line 216: Change “hemophagic” to “hemophagocytic”

This change has been made as requested

Lines 216-219: This long sentence is hard to read. Consider making 2 sentences.

This change has been made as requested.

Line 231: Change “40 samples” to “40 dogs” and change “diagnosis” to “enrollment” since some cases were relapsed. Make this change throughout the manuscript.

This change has been made as requested.

Line 233: Change “which” to “these dogs”

This change has been made as requested

Line 236-237: This statement that steroids may falsely decrease H3.1 plasma levels is new information and affects the reader’s interpretation of the data significantly. Consider just reporting numbers of dogs on steroids here and then include information about why and to what degree this may occur in discussion and/or intro.

This statement has been deleted from the results here and the discussion expanded in the discussion section.

Line 281: The readers need more information about exactly what “best response” was in these dogs and how it correlated with H3.1 concentration. For example, how many dogs achieved CR, PR, SD, and PD and how did their H3.1 concentrations differ from dogs with SD or PD?

This has been addressed as requested in lines 324-330.

Lines 289-293: Please be more clear as to what exact statistical comparison you are reporting here and throughout the manuscript. For example, “There was no significant difference in time to best response based on H3.1 nucleosome results” or similar.

We have tried to clarify this throughout the manuscript- this particular section has been updated in lines 341-350.

Lines 316-320: Most of this description of methodology should be in the methods section of the manuscript.

This change has been addressed as requested. 

Line 335: Consider listing total number of samples collected at beginning of results section along with total number of dogs.

The total number of samples collected has been added to the beginning of the results section. 

Line 351: Consider adding the word “obvious” between “no” and “concomitant diseases” since it is impossible to rule out undiagnosed conditions

This change has been made as requested. 

Lines 352-359: It is difficult to tell which of these dogs were in which groups. Consider starting by dividing all of these collections into major categories (elevated neutrophils and concomitant conditions, normal neutrophils and concomitant conditions, normal neutrophils no known concomitant conditions). Then discuss subgroups as appropriate.

This section has been clarified as requested. 

Case Examples: I will defer to the editor, but suggest that these case examples may be most appropriate in supplemental materials and not in the main text of the manuscript.

These cases have been to supplemental information. 

Line 366: It is not clear when this dog was actually enrolled in the study.

 All data included here is from first relapse. We have tried to clarify in the text.

Line 401: Change “great response” to “complete remission” or similar

This change has been made as requested.

Discussion:

Line 420: Your statement about nucleosome levels being associated with clinical remission may be true, but readers have not seen that specific data. You show lower H3.1 levels at “best response” but do not define that response or associate those responses with H3.1. Please clearly report comparisons between remission status using RECIST criteria and H3.1 concentrations in the results.

We have now tried to define ‘best response’ better throughout the manuscript using RECIST criteria. 

Line 423: Clarify that higher nucleosome levels “before treatment” are inversely correlated with survival (or similar).

This has been clarified in lines 546-549.

Lines 446-448: I assume that dogs in the trial had to have gross disease to enroll, so it seems that dogs out of remission or not in complete remission should still have elevated H3.1 levels. Are there other reasons steroids could affect H3.1 levels? A false negative results is also possible and should be discussed here.

Additional information about false negatives have been included in lines 611-615

Lines 462-464: This last sentence is confusing because of its placement after a discussion about human malignancies. State whether you are referring do your dog data or human data. If dog data, consider additional discussion interpreting this result or delete from this part of the discussion.

The authors meant this as a general statement, agnostic of species. This has been clarified in the text. 

Line 470: More discussion about the role of fasting in accuracy of H3.1 testing is warranted.

This has been addressed in lines 606-608

Line 488: How often did you see H3.1 level increases preceeding relapse? How long before relapse was clinically evident? This information would be very helpful for clinical veterinarians to know.

This information has been restated and expanded upon in lines 630-634. 

Line 489: Do you mean one time elevated concentrations? Concentrations that returned to normal without intervention? Please clarify what you mean here.

This was addressed in lines 654-662 as requested. 

Line 490-491: Because you did not report PPV, reword this sentence to be clear that this statement is a clinical interpretation of the data and not a statistical result.

This has been addressed as requested. 

Discussion general: The discussion adequately discusses the possibility of false positive H3.1 results, but does not adequately discuss the possibility of or causes of false negative results. Please include more discussion of this and how it could affect clinical interpretation. Additionally, there appears to be some inconsistency in the interpretation of gray zone results in the manuscript. Discussion of how to interpret a gray zone result is warranted.

This has been addressed as requested. 

Table 1:

Clarify footnote to state which group you compared the non-indolent T-cell LSA dogs PFS to.

 The footnote states that dogs with non-indolent T cell lymphoma were compared to dogs with B cell lymphoma. Please let me know if additional clarification is needed. 

Be consistent with reporting mean or median H3.1 concentrations in groups. If reporting medians, include range. If reporting means, include SD.

 Consistency with medians were applied across H3.1 groups, ranges were added for all median values and SD was added for all mean groups. 

Adding reference range for H3.1 concentration in the legend for table 1 would aid readers in interpretation.

Clarify whether number labeled as PFS and OST are medians. Also report range.

 This change has been made as requested

The last column with censoring and individual case information does not add to the table and should be removed.

 This column has been removed as requested. 

Table 2:

Be consistent with reporting mean or median H3.1 concentrations in groups. If reporting medians, include range. If reporting means, include SD.

Clarify whether number listed for PFS and OST is a median. Also report range.

 This change has been made as requested. 

The last column with censoring and individual case information does not add to the table and should be removed.

 This column has been deleted.

Only one significant digit need be included when reporting p-values.

 This change has been made as requested. 

Clarify footnotes to state which groups these cases were compared to.

 Additional information has been added for clarity. 

Figure 2.

Specify that these data are from the time of study enrollment

This change has been made as requested. 

Figure 3.

This figure would make more sense if best response data was between Diagnosis (enrollment) and PD since that would happen chronologically

This change has been made as requested. 

Figure 4. Are these data just LSA dogs or all dogs? Please clarify in the legend.

This has been clarified in the legend.

Figures 7-9: Considering moving with text to supplemental materials.

These have been moved to supplemental materials.

---

## [Decision Letter · Decision Letter 1]

4 Apr 2023

PONE-D-23-02803R1Monitoring plasma nucleosome concentrations to measure disease response and progression in dogs with hematopoietic malignancies.PLOS ONE

Dear Dr. Wilson-Robles,

Thank you for submitting your manuscript to PLOS ONE. After careful consideration, we feel that it has merit but does not fully meet PLOS ONE’s publication criteria as it currently stands. Therefore, we invite you to submit a revised version of the manuscript that addresses the points raised during the review process.

Please address the Reviewer's additional minor comments. ==============================

We look forward to receiving your revised manuscript.

Kind regards,

Douglas H. Thamm, V.M.D.

Academic Editor

PLOS ONE

Journal Requirements:

Reviewers' comments:

Reviewer's Responses to Questions

**Comments to the Author**

1. If the authors have adequately addressed your comments raised in a previous round of review and you feel that this manuscript is now acceptable for publication, you may indicate that here to bypass the “Comments to the Author” section, enter your conflict of interest statement in the “Confidential to Editor” section, and submit your "Accept" recommendation.

Reviewer #1: All comments have been addressed

Reviewer #2: (No Response)

2. Is the manuscript technically sound, and do the data support the conclusions?

Reviewer #1: Yes

Reviewer #2: Yes

3. Has the statistical analysis been performed appropriately and rigorously? 

Reviewer #1: Yes

Reviewer #2: Yes

4. Have the authors made all data underlying the findings in their manuscript fully available?

Reviewer #1: Yes

Reviewer #2: Yes

5. Is the manuscript presented in an intelligible fashion and written in standard English?

Reviewer #1: Yes

Reviewer #2: Yes

6. Review Comments to the Author

Reviewer #1: (No Response)

Reviewer #2: Thank you for your thorough responses to my previous comments. Only a few areas of clarification remain necessary as listed below.

Methods:

Add a brief statement in the methods section about what results for the Ni.Q assay were considered normal, grey zone, and elevated and/orr reference figure 1 for interpretation of results.

Results:

Lines 318-323: I assume nucleosome concentrations between dogs with CR vs PR vs SD are not statistically different. Please state that result here.

Table 3:

With the added data provided in lines 318-323, it is confusing to compare with the data in table 3. I see that the median nucleosome concentrations at SD and PR are 25.7 and 21.8, respectively, but table 3 shows SD and PR dogs with a mean of 50.51. Ideally there would be more consistency in reporting of means vs medians depending on whether data is normal or not.

Lines 362-364: Please state the statistical analysis you ran to come to this conclusion. Readers want to know which of the groups are significantly different.

7. PLOS authors have the option to publish the peer review history of their article (what does this mean?). If published, this will include your full peer review and any attached files.

Reviewer #1: No

Reviewer #2: No

---

## [Author Response · Author response to Decision Letter 1]

11 Apr 2023

6. Review Comments to the Author

Reviewer #1: (No Response)

Reviewer #2: Thank you for your thorough responses to my previous comments. Only a few areas of clarification remain necessary as listed below.

Methods:

Add a brief statement in the methods section about what results for the Ni.Q assay were considered normal, grey zone, and elevated and/or reference figure 1 for interpretation of results.

This has been added in lines 166-169.

Results:

Lines 318-323: I assume nucleosome concentrations between dogs with CR vs PR vs SD are not statistically different. Please state that result here.

This has been added to lines 323-325. 

Table 3:

With the added data provided in lines 318-323, it is confusing to compare with the data in table 3. I see that the median nucleosome concentrations at SD and PR are 25.7 and 21.8, respectively, but table 3 shows SD and PR dogs with a mean of 50.51. Ideally there would be more consistency in reporting of means vs medians depending on whether data is normal or not.

Thank you for catching this error. This has been corrected to 23.85 in table 5. 

Lines 362-364: Please state the statistical analysis you ran to come to this conclusion. Readers want to know which of the groups are significantly different.

This information has been added to the last paragraph in the methods section.

---

## [Editor Report · Decision Letter 2]

13 Apr 2023

Monitoring plasma nucleosome concentrations to measure disease response and progression in dogs with hematopoietic malignancies.

PONE-D-23-02803R2

Dear Dr. Wilson-Robles,

We’re pleased to inform you that your manuscript has been judged scientifically suitable for publication and will be formally accepted for publication once it meets all outstanding technical requirements.

Kind regards,

Douglas H. Thamm, V.M.D.

Academic Editor

PLOS ONE
---

## [Editor Report · Acceptance letter]

17 Apr 2023

PONE-D-23-02803R2 

Monitoring plasma nucleosome concentrations to measure disease response and progression in dogs with hematopoietic malignancies. 

Dear Dr. Wilson-Robles:

I'm pleased to inform you that your manuscript has been deemed suitable for publication in PLOS ONE. Congratulations! Your manuscript is now with our production department. 

Kind regards, 

on behalf of

Dr. Douglas H. Thamm 

Academic Editor

PLOS ONE